# Integrative Taxonomy of *Armeria arenaria* (Plumbaginaceae), with a Special Focus on the Putative Subspecies Endemic to the Apennines

**DOI:** 10.3390/biology11071060

**Published:** 2022-07-14

**Authors:** Manuel Tiburtini, Giovanni Astuti, Fabrizio Bartolucci, Gabriele Casazza, Lucia Varaldo, Daniele De Luca, Maria Vittoria Bottigliero, Gianluigi Bacchetta, Marco Porceddu, Gianniantonio Domina, Simone Orsenigo, Lorenzo Peruzzi

**Affiliations:** 1Department of Biology, University of Pisa, 56126 Pisa, Italy; lorenzo.peruzzi@unipi.it; 2Botanic Garden and Museum, University of Pisa, 56126 Pisa, Italy; giovanni.astuti@unipi.it; 3Floristic Research Center of the Apennine, University of Camerino—Gran Sasso Laga National Park, San Colombo, 67021 Barisciano, Italy; fabrizio.bartolucci@unicam.it; 4Department for the Earth, Environment and Life Sciences (DISTAV), University of Genoa, 16132 Genoa, Italy; gabriele.casazza@unige.it (G.C.); lucia.varaldo@edu.unige.it (L.V.); 5Department of Biology, University of Naples Federico II, 80100 Naples, Italy; daniele.deluca@unina.it (D.D.L.); ma.bottigliero@studenti.unina.it (M.V.B.); 6Centre for Conservation of Biodiversity (CCB), Department of Life and Environmental Sciences, University of Cagliari, 09123 Cagliari, Italy; bacchet@unica.it (G.B.); porceddu.marco@unica.it (M.P.); 7Department of Agricultural, Food and Forest Sciences, University of Palermo, 90128 Palermo, Italy; gianniantonio.domina@unipa.it; 8Department of Earth and Environmental Science, University of Pavia, Via S. Epifanio 14, 27100 Pavia, Italy; simone.orsenigo@unipv.it

**Keywords:** endemism, morphometrics, image analysis, molecular analysis, niche similarity, nomenclature

## Abstract

**Simple Summary:**

*Armeria arenaria* is a highly variable Western European species, for which three subspecies are recorded in Italy. *Armeria arenaria* subsp. *arenaria* has been reported from Northern Italy, while *A. arenaria* subsp. *marginata* and *A. arenaria* subsp. *apennina* are considered endemic to the Apennines. The taxonomic value of these two latter taxa is unclear and the actual occurrence of *A. arenaria* s.str. in Italy has never been addressed. Following an integrated taxonomic approach, in this study we show that all the Italian records of *A. arenaria* s.str. should be actually referred to *A. arenaria* subsp. *praecox* and that only one Northern Apennine endemic taxon can be recognized, namely, *A. arenaria* subsp. *marginata*.

**Abstract:**

Three subspecies of *Armeria arenaria* are reported from Italy, two of which are considered endemic to the Apennines. The taxonomic value of these two taxa (*A. arenaria* subsp. *marginata* and *A. arenaria* subsp. *apennina*) is unclear. Moreover, the relationships between *A. arenaria* subsp. *praecox* and Northern Italian populations—currently ascribed to *A. arenaria* subsp. *arenaria*—have never been addressed. Accordingly, we used an integrated taxonomic approach, including morphometry, seed morpho–colorimetry, karyology, molecular systematics (*psbA–trnH*, *trnQ–rps16*, *trnF–trnL*, *trnL–rpl32*, and *ITS* region), and comparative niche analysis. According to our results, French–Northern Italian populations are clearly distinct from Apennine populations. In the first group, there is evidence which allows the recognition of *A. arenaria* s.str. (not occurring in Italy) and *A. arenaria* subsp. *praecox*. In the second group, the two putative taxa endemic to the Northern Apennines cannot be separated, so a single subspecies is here recognized: *A. arenaria* subsp. *marginata.*

## 1. Introduction

Most of our biological knowledge of plant diversity comes from the foundations laid by alpha taxonomy, which played a crucial role in discovering and documenting plant diversity around the world. Nevertheless, although science is progressing, taxonomists seem to struggle to keep pace with novel methods and approaches. Indeed, hundreds of putative new species are described annually [1], but most of them are still described on qualitative grounds. In such approaches, the information that a taxonomist collects to shape his/her idea about the “species” in question is often obscure [2], so that biases [3] in taxonomist decisions [4] can dramatically affect taxonomic treatment. For instance, the number of species in the genus *Armeria* varies dramatically under different taxonomic circumscriptions elaborated by different taxonomists [5,6,7]. The subjectiveness of these processes may have contributed to what has been called taxonomic anarchy [8]. Integrated taxonomic approaches aim to address this problem with the consilience principle [2], according to which multiple and complementary approaches (morphology, phylogenetics, cytology, etc.) [9] are used to try falsifying taxonomic hypotheses in a Popperian sense [10]. This represents a step towards an omega taxonomy [11,12] that needs the integration of different skills [13,14].

The genus *Armeria* Willd. (Plumbaginaceae, Limonioideae) includes up to 95 accepted, mostly Holarctic, perennial species [15]. In Italy, the current knowledge on the taxonomy and systematics of this genus is largely derived from traditional alpha–taxonomic revisions [16,17], which indicate the existence of 23 taxa in 18 species [18]. However, the taxonomic value of some of these taxa is still debated [18], and the picture is further complicated by the fact that species boundaries within *Armeria* are difficult to establish [6,19] and weak [20,21,22]. In this scenario, it has been demonstrated that homoploid hybrid speciation [21] can play a crucial role in the emergence of new species [23,24], given that all species tested so far are diploid, with 2*n* = 2*x* = 18 chromosomes [16,25,26]. The use of nrDNA and (maternally inherited) cpDNA markers helps to elucidate the phylogenetic relationships even under hybridization scenarios [27,28]. *Armeria arenaria* (Pers.) F. Dietr. complex currently includes 13 subspecies in its whole range [29] and, according to Arrigoni [17], three subspecies occur in Italy: *A. arenaria* subsp. *arenaria*, distributed across the Central–Western Alps [18]; *A. arenaria* subsp. *apennina* Arrigoni, endemic to the Tuscan–Aemilian Apennines; and *A. arenaria* subsp. *marginata* (Levier) Arrigoni, also endemic to the Northern and up to the Central Apennines. *Armeria arenaria* subsp. *praecox* (Jord.) Kerguélen ex Greuter, Burdet & G. Long, described from south–eastern France, is reported as doubtfully occurring in Italy. Arrigoni [17] considers *A. arenaria* subsp. *apennina* as intermediate between *A. arenaria* s.str. and *A. arenaria* subsp. *marginata*. The same author [17] also claims that there is a series of unclear intermediate forms distinguished by the transition of some putatively diagnostic character states. However, the circumscription of these subspecies is based only on a qualitative morphological approach. All these factors led to the consideration of *A. arenaria* subsp. *apennina* and *A. arenaria* subsp. *marginata* as two subspecies of uncertain taxonomic value [18].

For these reasons, there is need to use an integrated approach to address the taxonomy [9] of these putative subspecies. To achieve a sound taxonomic circumscription, we performed morphometric analyses, including living populations from type localities, complemented by seed morpho–colorimetry, karyotype asymmetry estimation, molecular systematics, and comparative niche analysis (for similar integrative approaches, see [27,28]). In this study we aim: (1) to test the current taxonomic circumscription; (2) to verify the occurrence in Italy of *A. arenaria* subsp. *praecox*; and (3) to clarify the nomenclature of the group.

## 2. Materials and Methods

### 2.1. Sampling

In total, we selected 12 populations (Table 1) across the Northern Apennines and Western Alps up to Central France. The populations studied were selected based on three criteria: (1) to include all the type localities of the four taxa putatively occurring in Italy (FB, LA, LL, and MB—acronyms as in Table 1); (2) to include other populations explicitly cited in [19]: AA, BO, BR, MC, MP, and TV; and (3) to also include a lowland (GA) and the easternmost (PS) populations in Italy.

For each population, about 20 flowering individuals were sampled. The number of flowering scapes was counted in the field, whereas pictures were taken to assess the colour of the flowers and involucres of each plant. In total, 229 specimens were collected, and herbarium vouchers were prepared. All vouchers are stored at Herbarium Horti Botanici Pisani (PI), and high–resolution images are freely available for consultation at http://erbario.unipi.it/, accessed on 10 July 2022 (codes in Table 1). Concerning molecular systematics, dried leaves were picked from a subset of three individuals for each population and put in a paper bag with silica gel. Ripe fruits were also collected from the same populations. Seeds were dried at room temperature for two months and cleaned in the Germplasm Bank, Department of Biology of the University of Pisa, using sieves and Agriculex CB–1 Column Seed Cleaner complemented by manual cleaning.

### 2.2. Morphometric Analysis

In total, 49 qualitative and quantitative morphological characters (Table 2, see also Appendix A for details concerning the calyx) were studied, with a resulting dataset of 223 individuals × 49 variables. Macroscopic measures were taken with a digital calliper (error ± 0.1 mm), whilst microscopic and calyx measurements [30] (Table 2 and Appendix A) were taken through bar–scaled pictures with a Fiji 2.1.0 [31]. To provide a more objective means of counting the number of leaf veins, free–hand transversal sections of leaves were prepared. We considered as a “vein” each fascicule composed of the xylem and phloem surrounded by sclerenchyma. The anatomy of summer leaves was surveyed under a Leitz Diaplan light microscope at 40× (Appendix A). We considered as “involucral bracts” those from the capitulum involucre, as “spikelet bracts“ those subtending each spikelet, and as “bracteoles” those under each flower. To take into account the internal variability of the capitulum, we measured a spikelet collected from the middle of the capitulum (“inner spikelet”) and a spikelet in contact with the inner involucral bract (“outer spikelet”).

All statistical analyses were conducted in R Studio (version 3.6.2) [32]. To test the suitability of the data for factor analysis, the Kaiser–Meyer–Olkin test (MSA = 0.86, *psych* package [33] and Bartlett sphericity test (*p* < 0.001, *REdaS* package [34]) were performed successfully on the correlation matrix. Since there were mixed variables, Gower distance in the *FD* package [35] with Podani correction [36] was used, whilst Cailliez correction [37] was applied due to the violation of the triangle inequality (i.e., the matrix was not Euclidean). On such a dissimilarity matrix, Principal Coordinate Analysis (PCoA) in the *ape* package [38] was used to explore the dataset. Graphs were plotted with the *ggplot2* package [39]. One–way ANOSIM in PAST (version 4.09) [40] was used to test the null hypothesis of no difference between groups in the Gower dissimilarity matrix. To test the current taxonomic hypothesis and other alternative groupings based on our results, we applied jackknifed Linear Discriminant Analysis (LDA) in the *MASS* (for plotting) and *Predpsych* (to obtain the confusion matrix [41]) packages. Qualitative variables were converted into numbers with integer encoding. Using the *PredPsych* package, Cohen’s Kappa coefficient was estimated for each grouping hypothesis. K coefficient is a measure of how the classification results compare with the values assigned and is generally thought to be a more robust measure than simple percentage agreement calculation, since it considers the possibility of agreement occurring by chance [42]. It ranges from 0 to 1 and K values greater than 0.75 may be taken to represent excellent agreement beyond chance [43].

Each character was statistically tested. For all the quantitative characters, normality was tested with the Shapiro–Wilk test. Normal and non–normal data were checked for homoscedasticity with the Bartlett test and Levene–Brown–Forsythe test, respectively. After checking statistical assumptions, normal and log–normal quantitative characters were compared among groups with one–way ANOVA and the post hoc Tukey–Kramer test (homoscedastic data) or Welch’s ANOVA and the post hoc Games–Howell test (heteroscedastic data). On the contrary, non–normal characters were tested with the Kruskal–Wallis and multiple comparisons test of Wilcoxon–Mann–Whitney (homoscedastic data) or a permutation test implemented using the pairwisePermutationTest function of the *rcompanion* package [44] (heteroscedastic data). To control the family–wise error rate of multiple comparisons, Holm’s correction was applied to all the tests. Qualitative nominal and ordinal characters were tested with the R function pairwiseNominalIndependence and pairwiseOrdinalIndependence based on Fisher’s exact test and implemented in the *rcompanion* package [44]. All the tests were considered significant with α< 0.01. The number of statistically significant differences for a variable among population pairs was counted and a pairwise triangular matrix was built. Descriptive statistics for each group were calculated using the describeBy function in the *psych* package [33].

### 2.3. Seed Morpho–Colorimetric Analysis

For a sample of 100 seeds per accession (cleaned from the fruiting calyx and the membranous pericarp), digital images were acquired using a flatbed scanner (Epson Perfection V550) with a digital resolution of 1200 dpi. When an accession had fewer than 100 seeds, the analysis was carried out on the whole batch available. The system worked with 2D images: seeds were randomly disposed on the scanner tray, so that they did not touch one another, and covered using a box with white paper followed by a box with black paper to avoid interference from environmental light. The images were processed using the software package ImageJ (version 1.52b) (Available online: http://rsb.info.nih.gov/ij. (accessed on 11 March 2022), and the descriptors of seed–size, shape, and colour features were measured and analysed. A plugin, Particles8 [45] (Available online: https://blog.bham.ac.uk/intellimic/g-landini-software/. (accessed on 11 March 2022), was used to measure 20 colorimetric and 26 morphometric features. This plugin was further enhanced by adding algorithms that can compute the Elliptic Fourier Descriptors (EFDs) for each analysed seed, thus increasing the number of independent variables. Following Terral et al. [46] and Sarigu et al. [47], to minimize measurement errors and optimize the efficiency of shape reconstruction, 20 harmonics were used to define the seed boundaries, obtaining 78 additional variables that were useful to discriminate between the studied seeds. In total, 124 morphometric and colorimetric characters were measured for each seed [48]. Statistical analyses were performed with the software SPSS release 16 (SPSS 16.0 for Windows; SPSS Inc., Chicago, IL, USA) by applying stepwise Linear Discriminant Analysis (LDA).

### 2.4. Karyological Analysis

Seeds were germinated in Petri dishes with 1% agar at 25 °C in an alternating 12/12 h dark/light photoperiod. After about 4 days, radicles emerged, and seedlings were removed from the seed incubator and kept at 4 °C for 24 h in a fridge, then we followed the Feulgen staining protocol. Root tips were pre–treated with 0.4% colchicine for 3 h and then fixed in Carnoy fixative solution for 1 h. After hydrolysis in HCl 1 N at 60 °C for 8 min, the root tips were stained in leuco–basic fuchsine for 2 h; root tips were squashed in a solution of aceto–orcein on a microscope slide.

Chromosomes were observed with a Leitz Diaplan microscope at 100× and pictures were taken with a Leica MC–170HD camera using Leica LAS–EZ 3.0 imaging software. At least four good metaphase plates were measured for each population. Lastly, chromosome numbers and karyological variables, such as THL (Total Haploid Length), M_CA_ (Mean Centromeric Asymmetry), CV_CL_ (Coefficient of Variation of Chromosome Length), and CV_CI_ (Coefficient of Variation of Centromeric Index) were obtained from each plate with MATO 1.1 (version 20210101) [49]. Since all the karyological variables were normal and homoscedastic, they were statistically tested with One–way ANOVA and the post hoc Tukey–Kramer test for more than 3 groups or with two sample *t*–tests when comparing 2 groups.

### 2.5. DNA Extraction and Molecular Systematics

Total DNA was extracted using the GeneAll^®^ Exgene™ Plant SV mini kit (GeneAll Biotechnology, Seoul, Korea), following the manufacturer’s protocol for dried material. About 25 mg of leaf tissue was ground to powder using the Mixer Mill 300 (Retsch^®^, Verder Scientific, Haan, Germany). The quality and quantity of extracted DNA was evaluated by 0.8% gel electrophoresis using the high–molecular weight marker HyperLadder™ 1 Kb (Bioline, Meridian Bioscience, Cincinnati, OH, USA). The internal transcribed spacers *ITS1* and *ITS2(+5.8S*) and four chloroplast intergenic spacers (*trnF–trnL*, *trnH–psbA*, *trnL–rpl32*, *trnQ–rps16*) were amplified in a final volume of 25 µL containing: 10 ng DNA, 2X Kodaq PCR MasterMix (ABM^®^, Richmond, BC, Canada), 400 nM forward and reverse primers, and water to volume. The list of primers [50,51,52] and PCR conditions is reported in Appendix A. Amplification products were visualized by 1.5% gel electrophoresis and purified using 15–20% polyethylene glycol (PEG), according to the size of the fragment. The purified amplicons were sequenced at one (chloroplast markers) or both ends (ITS region) using the BrightDye^®^ Terminator Cycle Sequencing Kit (MCLAB, Harbor Way, San Francisco, CA, USA). Capillary electrophoresis was carried out using the Applied Biosystems^®^ 3130 Genetic Analyzer (Applied Biosystems, Thermo Fisher Scientific, Foster City, CA, USA). *ITS* sequences were submitted to GenBank (accession numbers: ON512680–ON512715), while the chloroplast intergenic spacers were submitted to DDBJ (*trnF–trnL*: LC710463–LC710498; *psbA–trnH*: LC710671–LC710706; *trnL–rpl32*: LC710707–LC710742; *trnQ–rps16*: LC710743–LC710778).

Sequences were visually inspected and aligned using the ClustalW algorithm [53] implemented in BioEdit (version 7.2.5) [54] with the default values. An incongruence length difference (ILD) test was carried out in Nona (version 2.0) [55], as a daughter process of Winclada (version 1.00.08) [56], to test the putative incongruence of nuclear and chloroplast partitions prior to combination; default values were used for the analysis. A nucleotide evolution model was calculated for each of the five sequenced regions using jModelTest (version 2.1.10) [57], and the best fitting model was chosen over the others using the Bayesian Information Criterion (BIC) [58]. A Bayesian phylogenetic tree was inferred in MrBayes (version 3.2.6) [59] in two simultaneous, independent runs with the following settings: 2,000,000 generations of MCMC sampling every 2000 generations, and four runs (three cold and one hot). Convergence and mixing were evaluated in Tracer (version 1.7.2) [60]. The consensus Bayesian tree was visualized in FigTree (version 1.4.2) [61]. The best evolution models were K80 [62] for *ITS* and F81 [63] for chloroplast markers.

### 2.6. Comparative Niche Analysis

Occurrence data for the studied taxa were retrieved directly in the field, from SILENE (French National Mediterranean Botanical Conservatory of Porquerolles) (Available online: http://flore.silene.eu/index.php?cont=accueil. accessed on 17 May 2022), GBIF (Available online: https://www.gbif.org. accessed on 17 May 2022), and Wikiplantbase #Italia [64], with a total of 496 points. To test for differentiation in environmental space, we represented and quantified niche overlap using the PCA–based method developed by Broennimann et al. [65]. The Schoener’s D index, which ranges from 0 (no overlap) to 1 (full overlap), was used to measure niche overlap [66]. We used niche similarity tests [67] to assess whether the ecological niches of the taxa were more similar than expected at random from their geographical ranges. Niche similarity tests compare the environmental conditions occupied by taxa, taking into account the environmental conditions that are available in the geographic area occupied by each taxon. Briefly, the observed climatic niche overlap between two taxa was compared, with the overlap measured between the niche of one taxon and the randomized niche of the other taxon. This randomized niche was obtained by randomly sampling occurrence points in buffer areas of 10 km around occurrences (the ‘background area’).

### 2.7. Nomenclature and Distribution

Currently accepted names, basionyms, and homotypic synonyms within *Armeria arenaria* and its subspecies studied here were taken from the Med–Checklist [68]. Information about the herbaria in which the original material could be stored was derived from [69]. Accordingly, we digitally examined the following herbaria: B, FI, L, LY, M, MPU, P, and SLA (herbarium acronyms follow Thiers [70]). Once we had elaborated the new taxonomic scheme, we used our identification key to assess the geographical distribution of the recognised taxa by checking the herbarium materials stored at ANC, APP, B, CAME, FI, HLUC, MJG, MW, P, and RO. This material was then georeferenced and used to build the map in Figure 6.

## 3. Results

### 3.1. Morphometry

The first two axes of the PCoA explain 49% of the total variance. Along the first axis, there is a clear separation of four Apennine populations (AA, LA, MB, and MC, on the right side of Figure 1). Hereafter, we will refer to this group of four populations as “marginatoid”. Another group uniting populations from northern Italy (BO, BR, GA, MP, PS, and TV) and France (FB and LL) emerged. We will refer to this group hereafter as “arenarioid” (on the left side of Figure 1). The MP population, initially attributed to *A. arenaria* subsp. *apennina*, clearly falls among arenarioid plants.

Along the second axis, the topotypical population of *A. arenaria* subsp. *arenaria* (FB) shows a slight separation from the other arenarioid populations (Figure 2). One–way ANOSIM showed that there was, indeed, a significant difference between FB and the rest of the arenarioid populations (BO, BR, GA, MP, PS, TV, and LL) (R = 0.6573, *p* = 0.001), confirming the separation shown along the second axis of PCoA. LDA performed on the current taxonomic hypothesis (Table 3) obtained an 87% correct classification and K = 0.8. The lowest value of sensibility was scored by *A. arenaria* subsp. *marginata* (77.7%), followed by *A. arenaria* subsp. *apennina* (85.4%). The percentage of correct classifications and K increased to 99% and 0.9696, respectively, when comparing arenarioid with marginatoid plants.

To further investigate the morphological variation within arenarioid plants, we carried out a pairwise comparison using univariate statistical analyses on single characters. Figure 2 shows that the highest number of pairwise differences (94) was found between FB and all the other arenarioid populations. The population that shows the second most number of differences is PS (72).

Accordingly, we set up two new alternative grouping hypotheses in both of which marginatoid plants (AA, LA, MB, and MC) were combined in a single group. In the first grouping hypothesis (I), we tested FB, together with all the Northern Italian populations, as belonging to the same taxon (as in the current taxonomic hypothesis) against the single population LL, which is the topotypical population of *A. arenaria* subsp. *praecox*. In the second grouping hypothesis (II), we tested LL as belonging to the same taxon as all the other Northern Italian arenarioid populations against the single population of FB (which corresponds to *A. arenaria* s.str.). The performance of LDA was 96% (K = 0.925) under grouping hypothesis I and 98% (K = 0.968) under grouping hypothesis II. The two most important qualitative characters are provided in Appendix A, whereas mean (± standard deviation) values of the quantitative morphological characters for each population are provided in Appendix A.

### 3.2. Seed Morpho–Colorimetry

The LDA performed on the current taxonomic hypothesis of a priori groups gave an overall cross–validated classification performance of 51.7% (Table 4). *Armeria arenaria* subsp. *marginata* showed the highest percentage of discrimination, with values of 71.5%, while the lowest (36.3 %) was detected in *A. arenaria* subsp. *apennina* (Table 4).

The second LDA, contrasting arenarioid and marginatoid plants, provides an overall percentage of 86% correct classification, with high discrimination performance for the two groups (Table 5).

According to two alternative grouping hypotheses derived from the morphometric analysis, FB was tested, together with all Northern Italian populations, as belonging to the same group, against the single population LL (hypothesis I), and LL was tested as belonging to the same group as all other Northern Italian arenarioid populations against the single population FB (hypothesis II). The discriminant analysis provided an overall percentage of classification of 77.3% and 84.4% for hypotheses I and II, respectively (Table 6, Figure 3). In hypothesis I, high discrimination performance was obtained for LL (78.8%) and marginatoid plants (81.3%). Concerning hypothesis II, higher performances, ranging from 91.0% (in FB) to 81.5% (in marginatoid plants), were detected.

### 3.3. Karyotype Structure and Asymmetry

All the studied populations were diploid, with 2*n* = 2*x* = 18 chromosomes. They showed medium–sized (4.68 ± 0.64 µm), mostly metacentric (48.6%) or submetacentric (50.7%), chromosomes (see also Appendix A). One–way ANOVA revealed that all four karyological indices showed no significant differences among the four subspecies as circumscribed according to the current taxonomic hypothesis. However, the arenarioid plants (*n* = 45) showed significantly lower M_CA_ (t = −4.52, df = 59, *p* < 0.001) and THL (t = −4, df = 59, *p* < 0.001) values when compared to marginatoid plants (*n* = 16). Lower, but not significantly different, values were also observed in CV_CL_ and CV_CI_. Mean (± standard deviation) values for the karyological indices at population level are provided in Appendix A.

Under grouping hypothesis I, keeping all the Italian arenarioid populations as *A. arenaria* subsp. *arenaria* and contrasting them with LL and with marginatoid plants, one–way ANOVA revealed that THL (F = 8.056; *p* < 0.001) and M_CA_ (F = 10.52, *p* < 0.001) were significantly different, but not CV_CL_ and CV_CI_. A post hoc Tukey–Kramer test showed that M_CA_ and THL values differed significantly (*p* < 0.001) between marginatoid and Italian arenarioid plants (+FB). In contrast, M_CA_ and THL were not significantly different between LL and the marginatoid group, or between the two arenarioid groups.

Under grouping hypothesis II, grouping all the Italian arenarioid populations with LL, contrasting them with FB and with marginatoid plants, One–way ANOVA revealed that THL (F = 8.158; *p* < 0.001), M_CA_ (F = 11.03; *p* < 0.001), and CV_CI_ (F = 5.221; *p* < 0.01) were significantly different among the three groups, while no difference was found in CV_CL_ values.

A post hoc Tukey test showed that M_CA_ and CV_CI_ values differed significantly between marginatoid plants and FB at *p* < 0.01, whereas THL differs significantly at *p* < 0.001 between marginatoid plants and all Italian arenarioid plants (+ LL) (Figure 4). In contrast, there was no significant difference between Italian arenarioid plants (+ LL) and FB in any of the studied karyological indices.

### 3.4. Molecular Systematics

The number of phylogenetically informative characters obtained from the amplification of the five markers was 36, corresponding to approximately 1.5% of the entire alignment. The markers that showed the highest number of informative characters were the intergenic spacers *trnL–rpl32* and *ITS*, with 13 and 11 phylogenetically informative characters (Appendix A). The results of the ILD test showed that all plastid markers were congruent (*p* > 0.05, Appendix A). On the contrary, increasing the number of replicates (up to 100), all the pairwise combinations were congruent except for *ITS* and *trn*L–*rpl*32, which turned incongruent at *p* = 0.01 (Appendix A). Indeed, removing the *trn*L–*rpl*32 marker, the *ITS* and the resulting concatenated plastid matrix become congruent (*p* = 0.09). However, since the topology of the concatenated trees with and without *trn*L–*rpl*32 were not in conflict, we decided to retain the full matrix, which was 2337 bp long.

The Bayesian concatenated consensus unrooted tree is shown in Figure 5. Arenarioid populations are split into two main clades but are collectively well distinct from marginatoid (+ arenarioid PS) populations. The former main clade is more variable and encompasses accessions from the French populations FB and LL (forming a clade), accessions from MP, a clade with the accessions from GA, TV, BO (the latter in a separate clade), as well as those from BR, which do not form a monophyletic group. The second main clade contains two clades and the accessions from PS with an unresolved position. Separate ITS and plastid phylogenies are provided in Appendix A.

### 3.5. Comparative Niche Analysis

Schoener’s D values were generally low, ranging from 0 to 0.208. In particular, *A. arenaria* subsp. *praecox* was the subspecies showing a niche that overlapped less with those of the other subspecies, according to the current taxonomic hypothesis (Table 7). The lack of significance in the similarity test indicated that the low niche overlap values were due to habitat availability in the background areas rather than an effect of habitat selection. Taken together, these results suggest differences in optimal niche positions without niche shift.

## 4. Discussion

All our results concur in highlighting that the current taxonomic hypothesis available for *Armeria arenaria* is no longer supported. Starting from the marginatoid plants, there is no morphometric support at all for distinguishing the two taxa as proposed by Arrigoni [17]. Moreover, the idea that *A. arenaria* subsp. *apennina* represents a taxon somehow intermediate between *A. arenaria* subsp. *marginata* and *A. arenaria* subsp. *arenaria* [17] is only supported by their climatic requirements. Nevertheless, it also should be noticed that the two putative marginatoid taxa show the highest values of niche overlap detected. There is no karyological difference between the two marginatoid taxa, but together they show higher M_CA_ and THL values with respect to the arenarioid plants. Phylogenetically, all marginatoid plants form a highly supported clade, in which the accessions of the two putative subspecies are intermingled. A single alpine arenarioid population (PS) is placed phylogenetically close to the marginatoid plants, suggesting that the genetic differentiation between arenarioid and marginatoid plants occurred only recently and may be derived from incomplete lineage sorting or gene flow. Despite this, morphological evidence fully places PS among arenarioid plants. Accordingly, we deem that the maintenance of the subspecific rank for marginatoid with respect to arenarioid plants is appropriate. Concerning arenarioid plants, they share a set of morphological and karyological features. Altogether, our data also support the maintenance of the subspecific rank for *Armeria arenaria* subsp. *praecox* with respect to *A. arenaria* subsp. *arenaria*, albeit with different circumscriptions, since all the Italian arenarioid populations agree much better with *A. arenaria* subsp. *praecox* than with *A. arenaria* subsp. *arenaria.* Indeed, from a morphometric point of view, the FB population (*A. arenaria* s.str.) shows the highest number of pairwise differences among all the other arenarioid populations, and it also shows the smallest seeds.

As a consequence, we exclude *Armeria arenaria* subsp. *arenaria* from the Italian flora, in favor of *A. arenaria* subsp. *praecox*, so that the range of the former subspecies is now reduced to Portugal, Spain, and France [68]. We cannot rule out that the range of that subspecies could be further narrowed in the future, given that this taxon is *“conceived as a mixed bag that includes the variability of the rest of the populations”* [25,71]*. Armeria arenaria* subsp. *praecox* has been only doubtfully recorded for Italy so far [17]. However, we clearly show that Italian arenarioid plants have a morphology highly overlapping that of the typical *A. arenaria* subsp. *praecox* (Figure 1). Indeed, the highest values of correct classification and K obtained by the discriminant analyses conducted for morphology and seed morpho–colorimetry were found when all the Italian arenarioid populations were grouped with LL and not with FB (which corresponds to the typical *A. arenaria* s.str.). Italian arenarioid populations are also phylogenetically more closely related to LL than to FB in the plastid tree (Appendix A). The possible occurrence in Italy of other subspecies occurring in Southern France can be excluded based on the comparison of our data with those published by Baumel et al. [72], Tison et al. [73], and Tison et De Foucault [74] (data not shown).

Geographically and climatically, the marginatoid plants from the Northern Apennines are replaced in the central–western alpine and perialpine areas by *A. arenaria* subsp. *praecox*, which is in turn replaced by *A. arenaria* subsp *arenaria* in Central–Northern France (Figure 6).

**Figure 6 biology-11-01060-f006:**
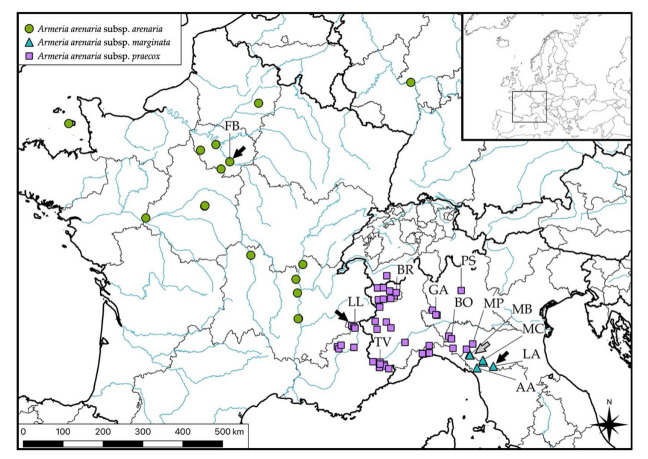
Distribution based on 81 herbarium specimens, including the localities sampled in this study, of *Armeria arenaria* subsp. *arenaria*, *A. arenaria* subsp. *marginata*, and *A. arenaria* subsp. *praecox*, as newly circumscribed. Solid arrows indicate type localities of the three taxa listed in the legend at the top–left corner on the map, whereas the crosshatched arrow indicates the type locality of *A. arenaria* subsp. *apennina*. Population codes as in Table 1.

## 5. Taxonomic and Nomenclatural Scheme

### 5.1. Identification Key

1.Leaves with flat apex, middle vein translucid or nearly so, veins (section!) usually ≤ 5, bracts of the inner spikelet 8.78 ± 0.95 mm long, capitula 18.3 ± 1.5 mm …….…….……...…….…….…...….……………...….***Armeria arenaria* subsp. *marginata***1.Leaves with cucullate apex, middle vein not translucid, veins (section!) usually > 5, bracts of the inner spikelet 6.67 ± 0.91 mm long, capitula 16.6 ± 2 mm wide……..……..……..……..…………………………...…………………………………... **2**2.Capitulum intermediate scales 3.87 ± 0.81 mm long; capitula 13.91 ± 1.5 mm wide; calyx 0.7 ± 0.15 mm wide, seeds 2.38 ± 0.07 mm long ……….……….……….………………………….…......***Armeria arenaria* subsp. *arenaria***2.Capitulum intermediate scales 5.98 ± 0.89 mm long; capitula 16.57 ± 2 mm wide; calyx 1 ± 0.16 mm wide, seeds 2.89 ± 0.25 mm long ……….……….……….……….……….…...…...…........***Armeria arenaria* subsp. *praecox***

### 5.2. Nomenclature and Distribution

***Armeria arenaria*** (Pers.) F.Dietr., Nachtr. Vollst. Lex. Gärtn. 1: 313 (1815) subsp. ***arenaria*** ≡ *Statice arenaria* Pers., Syn. Pl. 1: 332 (1805) ≡ *Armeria arenaria* (Pers.) Schult. in Roem. & Schult., Syst. Veg., ed. nov. (15), 6: 771 (1820) isonym ≡ *Armeria arenaria* (Pers.) Ebel, Armeriae Gen. Diss. 35 (1840) isonym.

Type: (neotype, here designated):—FRANCE. *Statice arenaria*, freq. rura Parisiis et alibi, s.d., *Persoon s.n.* (L2648462!).

In the protologue, a short diagnosis (“*caul. scapo longo, bract 2–3 capitulo longiorib., fol. linearib. rigidulis glabris*”), the habitat (“*in arenosis*”), and the provenance (“*Copiose prope Fontainebleau*”) are provided. No original material occurs at L (M. Scherrenberg, pers. comm.), where the Persoon’s herbarium and types are deposited [75].

Specimens seen. GERMANY. Gonsenheimer Wald bei Mainz, 8 July 1876, *A. Vigener* (FI!). BAILIWICK OF JERSEY. The Quennevais Jersey, 1860, *H.L.* (FI!). FRANCE. Sables aux Aulnois–sous–Laon (Aisne), 10 July 1873, *Favre* (FI!); Forêt de Rambouillet (Seine et Oise), sables au champ de manoeuvres le long de la route de Saint–Léger, 13 August 1932, *B. de Retz* (P05086601!); Prairies sèches, bois secs et clairs sur l’alluvion, près du Plessis–Piquet, aux environs de Paris, July 1841, *Kralik* (P05386780!); Nei dintorni de l’Hippodrome de la Solle, Fontainebleu (Seine–et–Marne) nelle schiarite sabbiose del bosco (WGS84: 48°26′13.9″ N 2°41′24.588″ E), 17 June 2020, *F. Losacco et M. Tiburtini* (PI56593–PI56615!); Buthiers (Seine–et–Marne) friches sablonneuses sur le coteau dominant la route de Malasherbes, 27 June 1943, *B. de Retz* (P05086604!); Gallia—La Sologne. In arenosis, 22 July 1924, *G. Lacaita* (FI!); La Sologne, entre Gien et Orléans: in arenosis, 22 July 1924, *G. Lacaita* (FI!); Saint-Nicolas, grande route de Thours (Tours), July 1845, *L. d’Espianay* (P00707605!); Moulis, sables de l’Allier, Jul 1890, *H. Bourdot* (FI!); Feillens près Mâcon, 7 July 1872, *H. Lareniq* (FI!); Arnas, par Villefranche (Rhone), France, in arenosis, 7 September 1876, *M. Gandoger* (FI!); Gall. Lyon, August 1900, *M. Gandoger* (FI!); Serras (Ardèche): prairies sablonneuses, 6 May 1878, *E. Chabert* (FI!).

***Armeria arenaria*** subsp. ***marginata*** (Levier) Arrigoni, Fl. Medit. 25 (Special Issue): 15 (2015) ≡ *Armeria majellensis* var. *marginata* Levier, Atti Soc. Tosc. Sci. Nat. Pisa Processi Verbali 6 (11 novembre 1888): 157 (1888) ≡ *Armeria marginata* (Levier) Bianchini, Giorn. Bot. Ital. n.s., 111: 49 (1977)

Type (lectotype, designated by [17]: 15):—ITALY. In monti Libro Aperto, Apennini Pistoriensis supra Boscolungo, 1700 m, July 1881, *Levier s.n.* (FI barcode FI002438!) = *Armeria arenaria* subsp. *apennina* Arrigoni, Fl. Medit. 25 (Special Issue): 13, Figure 1 (2015). Type (holotype): ITALY. Emilia–Romagna, Corniglio (Parma). Vaccinieti e rupi della cresta rocciosa tra il M.te Marmagna e M.te Braiola, m 1600–1800, substr. Arenaria, 21 July 1986, *Arrigoni, Foggi et Ricceri s.n.* (FI barcode FI007466!)

The two names were published simultaneously at subspecies level by Arrigoni [17] on November 20, 2015. We opt here for *Armeria arenaria* subsp. *marginata* as having priority over the competing *A. arenaria* subsp. *apennina* (Art. 11.5 of the ICN [76]).

Specimens seen. ITALY. Emilia–Romagna: Lago santo—Sotto il M. Orsaio versante parmigiano, 28 June 1902, *S. Sommier* (FI!); Vetta del Monte Cusna (Rescadore—Reggio Emilia) a 2100 m s.l.m. (WGS84: 44°17′17.538″ N 10°23′24.192″ E), 26 June 2020, *M. Tiburtini et S. Quitarrá* (PI54653–PI54672!); Monte Cusna—prati rocciosi della vetta, 10 August 1988*, B. Foggi et C. Ricceri* (FI!); Ligonchio M.te Prado. Cresta rocciosa tra lo sperone di Prado e la vetta Prati e Vaccinieti, Alt. 1955–2054, substrato: arenaria, 28 July 1987*, B. Foggi et C. Ricceri* (FI!); Vetta del Libro Aperto (Fiumalbo—Modena) a 1930 m s.l.m. (WGS84: 44°9′25,64″ N 10°42′45,078″ E), 25 June 2020, *M. Tiburtini et S. Quitarrá* (PI56573–PI56592!); Toscana: Salendo da Pracchiola al M. Orsaio Pascoli verso 1900 m, 28 June 1903, *S. Sommier* (FI!); *ibidem*, 1700 m, 28 June 1903, *S. Sommier* (FI!); *ibidem*, 1200 m, 28 June 1903, *S. Sommier* (FI!); Pascoli alpini al M. Orsaio presso la Foce di Catelea e la cima, 21 July 1838, *F. Parlatore* (FI!); Lungo il sentiero sulla Sella del Braiola (Lagdei, Parma) a circa 1600 m s.l.m. (WGS84: 44°24′4.512″ N 9°59′39.318″ E), 27 June 2020, *M. Tiburtini et S. Quitarrá* (PI53990–PI54009!); Appennino lucchese reggiano M. Prado erbosi su macigno esposti a sud vicino alla vetta alt. 2000 m, 20 August 1992, *E. Ferrarini* (FI!); Sommità di Monte Prado nelle Alpi di Mommio, July 1851, *F. Calderini* (FI!); Libro Aperto Appennino Pistoiese, 7 August 1898, *S. Sommier* (FI!); Alpi Apuane, Serenaia Minucciano sotto l’Orto di Donna, 28 May 1960, *B. Lanza* (FI!); *ibidem*, 1000 m, 28 May 1960, *B. Lanza* (FI!); Ambienti prativi rocciosi lungo il sentiero nei pressi del Masso del Gigante (Località Altare) sotto Foce di Cardeto, Alpi Apuane (Minucciano—Lucca) (WGS84: 44°7′26.98″ N 10°12′43.188″ E), 26 June 2020, *M. Tiburtini et S. Quitarrá* (PI54673–PI54686!).

***Armeria arenaria*** subsp. ***praecox*** (Jord.) Kerguélen ex Greuter, Burdet & G.Long, Med–Checkl. 4: 309 (1989) ≡ *Armeria praecox* Jord. In Boreau, Fl. Centre France, ed. 3, 2: 537 (1857) ≡ *Armeria arenaria* subsp. *praecox* (Jord.) Kerguélen, Lejeunia nov. ser., 120: 49 (1987), comb. inval. (Art. 41.5 of the ICN [72]).

Type (lectotype, here designated):— FRANCE. Hautes–Alpes, Monêtier–les–Bains, 1839, *Jordan s.n.* (LY barcode LY0421392!).

In the protologue, the name *Armeria praecox* is published in a note, reporting a short diagnosis and the provenance (“dans les Alpes”). Both the name (“*A. praecox* Jord.!”) and description are clearly attributed to Claude Thomas Alexis Jordan (Boreau, Fl. Centre France, ed. 3, 1: 12. 1857: “*Quant aux espèces que M. Jordan m’a communiquées avant de les avoir publiées, je me suis efforcé d’en saisir les caractères, et s’ils ne sont pas convenablement mis en lumière, c’est mon insuffisance seule qui devra être mise en cause*”). We traced one specimen at LY (barcode LY0421392), where Jordan’s herbarium and original material are preserved [75,77,78]. This specimen bears five parts of the same plant and the label “*Armeria praecox* Jord. | Hautes Alpes Monetier 1839 | (Herbier Jordan)”. LY0421392 is part of the original material, is congruent with the Boreau’s diagnosis, and is here designated as the lectotype for this name.

Specimens seen. FRANCE. Prairies de Serras—Ardèche, 8 May 1869, *Chabert* (FI!); Bords du le Mont du Lautaret, 23 July 1899, *P. Favre* (FI!); Le Lauzet (Hautes–Alpes): Lieux secs le long des chemins, 8 August 1875, *P. Favre* (FI!); Tra Le Lauzet e Le Monêtier–les–Bains (Hautes–Alpes) lungo i bordi della strada carrabile (WGS84: 44°58′48.588″ N 6°29′58.65″ E), 16 June 2020, *F. Losacco et M. Tiburtini* (PI55549–PI55568!); Col Bayard près Gaps: H.tes Alpes 1300 m, 23 June 1900, *L. Girod* (FI!); Le Roche–des–Arnauds près Gap, Jul 1886, *Serres* (FI!); La Freyssinouse (H.tes Alpes) 1000 m alt., 13 June 1898, *L. Girod* (FI!); Lieux incultes au Lauzet (H.tes Alpes), France, 2 August 1889, *E. Neyraut* (FI!); Terrains incultes au Lauzet—Hautes Alpes, 5 May 1883, *E. Neyraut* (FI!); Saint–Martin Vésubie, vallon de Salèses, 28 July 1910, *R. Pampanini* (FI!); Alpes de Tende, July 1843, *G.F. Reuter* (FI!). SWITZERLAND. Helvetia: Vallesia centralis in pratis saxosis aridis Vallis Hérens “Intericos la Sage et al. Forclas”, in consortio *Dianthus carthusianorum, Aster alpini, Galium borealis* etc. solo silic., 1700 m, 12 August 1926, *A. Remaud* (FI!). ITALY. Valle d’Aosta: Près de Saint Rémy en Aoste (Italy) pelouses sèches, bords des champs 1630 m, 2 July 1875, *F. Tripet* (FI!); Tra Ollomont e Valpelline 1000–1400 m, 25 June 1902, *L. Vaccari* (FI!); Saint Marcel valle inf. fino a Prabornaz, 7 August 1902, *L. Vaccari* (FI!); Cogne salita al M. Herban, 1500–2000 m, s.d., *L. Vaccari* (FI!); Valpelline et Oyace, 17 July 1914, *P. Bolzon* (FI!); Aosta tra Arpuille e Plau de Dian—1300–1500 m, 24 July 1899, *L. Vaccari* (FI!); Valle di Champorcher a 700 m, 1 June 1899, *L. Vaccari* (FI!); Balze su substrato ofiolitico e prati sfalciati circostanti, Castello di Graines (Brusson, Aosta) (WGS84: 45°44′17.4″ N 07°45′14.8″ E), 16 June 2020, *S. Orsenigo* (PI53970–PI53989!); Bassa Val d’Ayas, tra Barme e Carogne, sopra il castello di Verrés, in prato secco, 810 m, 23 May 2006, *M. Bovio* (FI!); Degioz—Valsavarenche Valle d’Aosta, 26 July 1935, *U. Losacco* (FI!). Piemonte: Ceresole Reale nei prati in fondo alla valle sotto la chiesa 1400 m, 25 July 1910, *L. Vaccari et Wileyk* (FI!); Regione Valensana, 20 May 1914, *P. Bolzon* (FI!); Mt. Musiné Piedmont, 21 May 1870, *A. Chamber* (FI!); Stupinigi, dans le bois Piedmont, April 1854, *A. Chevalier* (FI!); Perosa, erbosi sopra Pomaretto, 4 July 1937, *G. Negri* (FI!); Env. d’Alba et de Turin, 1868, *Borguais* (FI!); Laghi della Lavagnina (Ovada), s.d., *I. Vagge* (ANC6837); Vallone di San Bernoni–Bernolfo (Alpi Marittime), 23 July 1889, *A. Ferrari* (FI!); Vallone della Meris: erbosi aridi a 1500 m, Val Gesso (A.M.), 1 August 1961, *P.G. Bono* (FI!); Luoghi silvestri alla confluenza del torrente Vallasco nel Gesso, alle Terme di Valdieri m. 1370 frequentissima nei luoghi aridi attorno alle terme, 12 July 1897, *O. Boggiani* (FI!); Prati e affioramenti rocciosi lungo la mulattiera che porta al rifugio Valasco (Terme di Valdieri, Cuneo) (WGS84: 44°12′14.7″ N 07°15′54.0″ E), 29 June 2020, *S. Orsenigo* (PI55569–PI56573!); Alpi Marittime, Valle del Gesso Tra Entracque e San Giacomo, 11 August 1887, *T. Caruel* (FI!); Erbosi aridi in Vallone di M.te Colombo presso Prà del Bosur (1350–1400 m), Val Gesso (A.M.), 17 July 1961, *P.G. Bono* (FI!); Lombardia: Prati sfalciati (Salmezza, Bergamo) (WGS84: 45°46′57.3″ N 09°43′56.2″ E), 12 June 2020, *S. Orsenigo* (PI53950–PI53969!); Abbiategrasso Valle del Ticino = Siti sabbiosi secchi, 4 June 1895, *C. Camperio* (FI!); Radure aride, su substrato sabbioso/ghiaioso, Parco del Ticino (Gambolò, Pavia) (WGS84: 45°16′07″ N 8°57′39.7″ E), 26 May 2020, *S. Orsenigo* (PI54032–PI54049!); Zelata, Pavia, 1844, *L. Rota* (FI!); Prov. di Pavia Sassi Neri (Penice) serpentini 600–700 m, 1 August 1916, *Mafra* (FI!); Liguria: Monte Maggiorasca, s.d., *s.coll.* (FI!); Monte Beigua (Lig. occ.) alt. 1200 m, 26 July 1885, *N. Mezzana* (FI!); Appennino a Voltri, 2 July 1871, *Baglietto* (FI!); Arenzano M.te Tardia, 2 July 1871, *Grey* (FI!); Liguria—Varazze M. Sciguello 1160 m, 23 May 1928, *S*. *Fresino* (FI!); Emilia–Romagna: Su detrito e roccia, affioramento ofiolitico, Monte Tre Abati (Bobbio, Piacenza), 3 June 2020, *S. Orsenigo* (PI54050–PI54063!); Monte Prinzera (Fornovo di Taro, Parma) su serpentini (WGS84: 44°38′27.048″ N 10°5′1.58″ E), 12 June 2020, *G. Astuti et M. Tiburtini* (PI53010–PI54029!); Aemilia—Prov. di Parma, abunde in rupium fissuris montis Prinzera, solo siliceo, 20 May 1905, *P. Bolzon* (MW0786588!); Emilia—Appennino Parmense Val Taro Roccamurata in loc. Groppo di Gorro substr. Serpentinoso, 1 June 1980, *F. Roffi* (FI!).

## 6. Conclusions

In this work, we conducted an integrative taxonomic study of *Armeria arenaria* in Italy. On the basis of nomenclatural, morphometric, seed morpho–colorimetric, karyological, molecular, and comparative niche evidence we were able to demonstrate that the current taxonomic setting for this species is no longer supported. Specifically, we proved that *A. arenaria* subsp. *apennina* is a heterotypic synonym of *A. arenaria* subsp. *marginata* and that all the previous records of *A. arenaria* subsp. *arenaria* for Italy should be attributed to *A. arenaria* subsp. *praecox.* Finally, we also provided an identification key for dried herbarium specimens to facilitate the identification of these taxa. The same key was used to reconstruct the distribution of the three subspecies based on 81 herbarium specimens.

## Figures and Tables

**Figure 1 biology-11-01060-f001:**
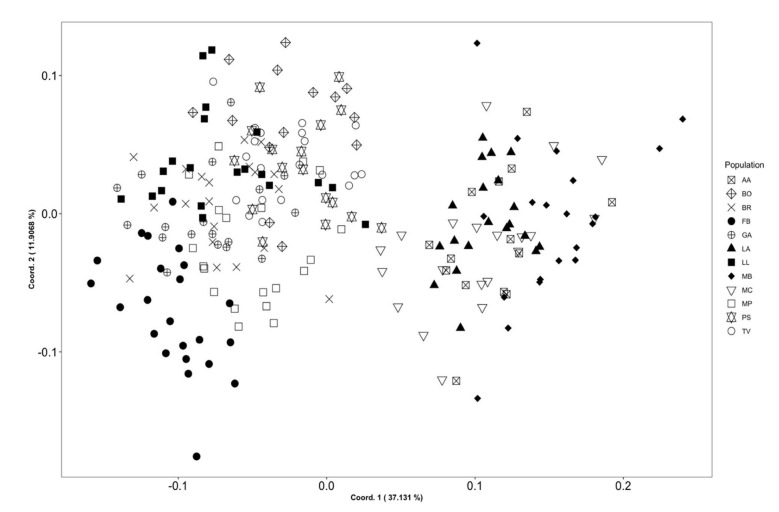
PCoA based on the 49 morphological characters measured in *Armeria arenaria* populations. Solid symbols represent individuals from type localities of the four taxa studied. AA = Apuan Alps, N Apennines; BO = Bobbio, N Apennines; BR = Brusson, Pennine Alps; FB = Fontainebleau, Île–de–France; GA = Gambolò, West Po Valley; LA = Libro Aperto; N Apennines; LL = Le Lauzet, Dauphiné Alps; MB = Marmagna–Braiola, N Apennines; MC = Monte Cusna, N Apennines; MP = Monte Prinzera, N Apennines; PS = Piana di Salmezza, Lombard Prealps; TV = Terme di Valdieri, Maritime Alps. Further population details are provided in Table 1 and Figure 6.

**Figure 2 biology-11-01060-f002:**
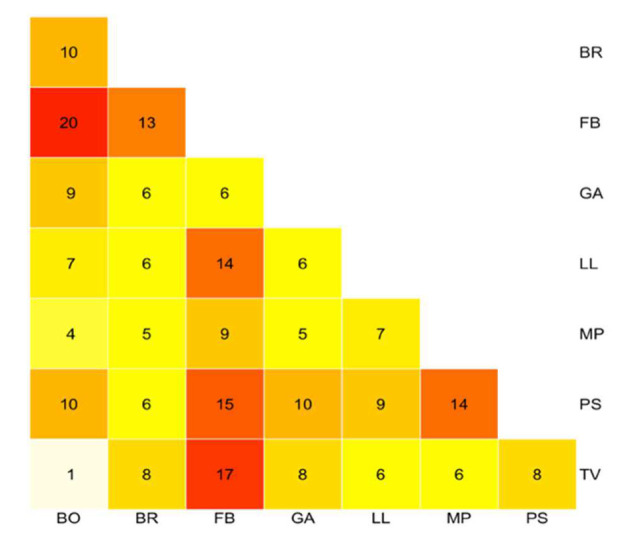
Heatmap of the pairwise comparisons of the 49 morphological characters for which we found statistically significant differences between population pairs in *Armeria arenaria*. Numbers inside the cells indicate the sum of statistically different characters. Colours are a function of the number of characters showing significant differences: whitish–yellow colours indicate that the pair is almost identical, whereas orange–red colours indicate that the pair shows several differences. AA = Apuan Alps, N Apennines; BO = Bobbio, N Apennines; BR = Brusson, Pennine Alps; FB = Fontainebleau, Île–de–France; GA = Gambolò, West Po Valley; LA = Libro Aperto; N Apennines; LL = Le Lauzet, Dauphiné Alps; MB = Marmagna–Braiola, N Apennines; MC = Monte Cusna, N Apennines; MP = Monte Prinzera, N Apennines; PS = Piana di Salmezza, Lombard Prealps; TV = Terme di Valdieri, Maritime Alps. Further population details are provided in Table 1 and Figure 6.

**Figure 3 biology-11-01060-f003:**
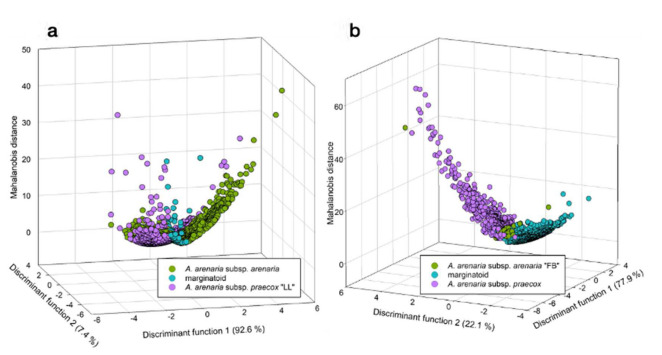
Graphical representation of the linear discriminant analysis (LDA) for the alternative grouping hypothesis for *Armeria arenaria* populations. (**a**) Grouping hypothesis I; (**b**) Grouping hypothesis II. FB = Fontainebleau, Île–de–France; LL = Le Lauzet, Dauphiné Alps.

**Figure 4 biology-11-01060-f004:**
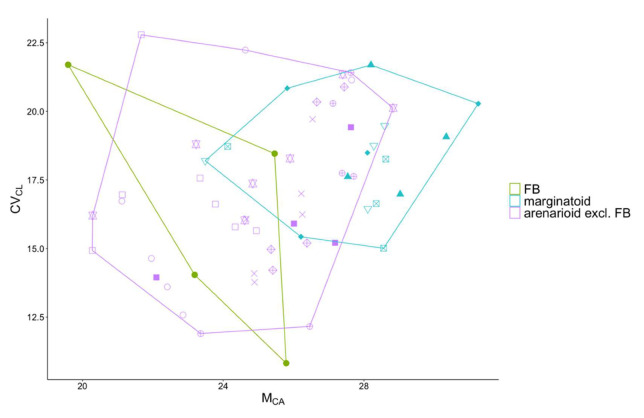
Scatterplot of the two karyotype asymmetry indices M_CA_ vs. CV_CL_ in *Armeria arenaria*. Accessions are enclosed by convex hulls according to grouping hypothesis II derived from the morphometric and seed morpho–colorimetric analysis, which sees LL as belonging to the same group as all the other Northern Italian arenarioid populations against the single population FB (which corresponds to *A. arenaria* s.str.). Symbols of populations as in Figure 1. FB = Fontainebleau, Île–de–France.

**Figure 5 biology-11-01060-f005:**
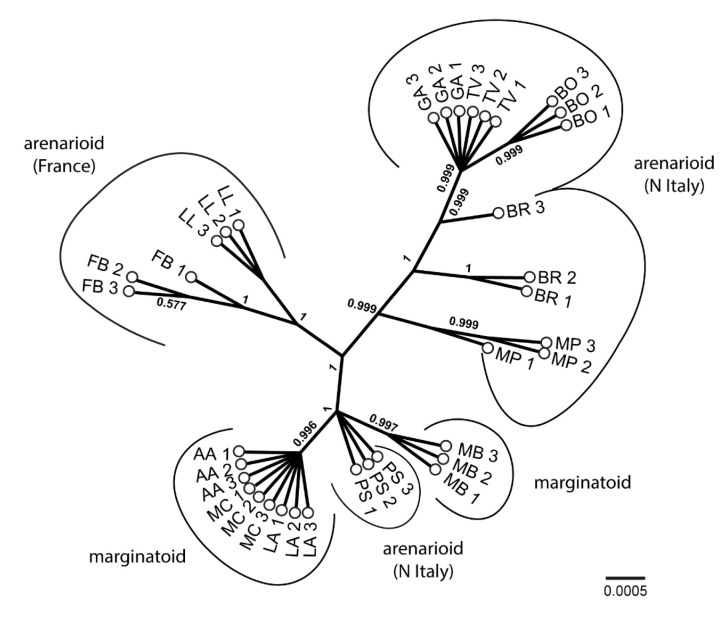
Bayesian unrooted consensus phylogenetic tree (concatenated dataset) of *Armeria arenaria* populations. AA = Apuan Alps, N Apennines; BO = Bobbio, N Apennines; BR = Brusson, Pennine Alps; FB = Fontainebleau, Île–de–France; GA = Gambolò, West Po Valley; LA = Libro Aperto; N Apennines; LL = Le Lauzet, Dauphiné Alps; MB = Marmagna–Braiola, N Apennines; MC = Monte Cusna, N Apennines; MP = Monte Prinzera, N Apennines; PS = Piana di Salmezza, Lombard Prealps; TV = Terme di Valdieri, Maritime Alps. Further population details are provided in Table 1 and Figure 6.

**Table 1 biology-11-01060-t001:** Taxa and populations of *Armeria arenaria* sampled in this study, according to the current taxonomic hypothesis [17]. “Code” corresponds to population acronyms used elsewhere in the manuscript. * = type locality. “Voucher” refers to the specimens stored at Herbarium Horti Botanici Pisani (PI) and freely available for consultation at http://erbario.unipi.it/, accessed on 10 July 2022. See also Figure 6 for the geographical localisation of the sampled populations.

Code	Current Taxonomic Hypothesis	Population	Voucher	*n*
MB *	*A. arenaria* subsp. *apennina*	Italy, Emilia–Romagna, between M. Marmagna & M. Braiola (WGS84: 44.401250 N, 9.994250 E)	*M. Tiburtini* et *S. Quitarrá*; 27 June 2020; PI53990–PI54009	20
MC	*A. arenaria* subsp. *apennina*	Italy, Emilia–Romagna, Monte Cusna (WGS84: 44.288194 N, 0.390055 E)	*M. Tiburtini* et *S. Quitarrá*; 26 June 2020; PI54653–PI54672	20
MP	*A. arenaria* subsp. *apennina*	Italy, Emilia–Romagna, Monte Prinzera (WGS84: 44.640833 N, 10.083772 E)	*M. Tiburtini* et *G. Astuti*; 12 June 2020; PI53010–PI54029	20
BO	*A. arenaria* subsp. *arenaria*	Italy, Emilia–Romagna, Monte Tre Abati, loc. Bobbio (WGS84: 44.752425 N, 9.436694 E)	*S. Orsenigo*; 3 June 2020; PI54050–PI54063	14
BR	*A. arenaria* subsp. *arenaria*	Italy, Val d’Aosta, Brusson (WGS84: 45.738166 N, 7.754111 E)	*S. Orsenigo*; 16 June 2020; PI53970–PI53989	20
FB *	*A. arenaria* subsp. *arenaria*	France, Île–de–France, Fontainebleau (WGS84: 48.437194 N, 2.690166 E)	*M. Tiburtini* et *F. Losacco*; 18 June 2020; PI56593–PI56615	23
GA	*A. arenaria* subsp. *arenaria*	Italy, Lombardia, Gambolò (WGS84: 45.268611 N, 8.961027 E)	*S. Orsenigo*; 26 May 2020; PI54032–PI54049	18
PS	*A. arenaria* subsp. *arenaria*	Italy, Lombardia, Piana di Salmezza (WGS84: 45.782583 N, 9.732277 E)	*S. Orsenigo*; 12 June 2020; PI53950–PI53969	20
TV	*A. arenaria* subsp. *arenaria*	Italy, Piemonte, Terme di Valdieri (WGS84: 44.204083 N, 7.265000 E)	*S. Orsenigo*; 26 June 2020; PI55569–PI56573	20
AA	*A. arenaria* subsp. *marginata*	Italy, Toscana, Alpi Apuane (WGS84: 44.124166 N, 10.212000 E)	*M. Tiburtini* et *S. Quitarrá*; 28 June 2020; PI54673–PI54686	14
LA *	*A. arenaria* subsp. *marginata*	Italy, Emilia–Romagna, Libro Aperto (WGS84: 44.157402 N, 10.712021 E)	*M. Tiburtini* et *S. Quitarrá*; 25 June 2020; PI56573–PI56592	20
LL *	*A. arenaria* subsp. *praecox*	France, Hautes Alpes, Le Lauzet (WGS84: 44.980166 N, 6.499611 E)	*M. Tiburtini* et *F. Losacco*; 16 June 2020; PI55549–PI55568	20

**Table 2 biology-11-01060-t002:** Morphological characters studied in *Armeria arenaria*. QC = quantitative continuous, QD = quantitative discrete, CN = nominal, BI = binary, CO = ordered factor.

Character Name	Description of the Character	Type	Tool
SCAB	Calyx awns scabrous (yes/no)	BI	Stereo
AWN	Awn presence on the calyx’s limb (yes/no)	BI	Stereo
CALYX_HAIRINESS	Calyx hairiness (holotrichous/pleurotrichous)	BI	Stereo
CALYX_VEINS	Number of calyx veins with hairs (10/5)	BI	Stereo
CAP_SHAPE	Shape of capitulum (hemispherical/subspherical)	BI	Stereo
DIMORP	Leaf dimorphism (yes/no)	BI	Stereo
INNER_SPI_BRACT_HAIR	Presence of hairs on inner spikelet bract (yes/no)	BI	Stereo
MAR_SUM_LEAF	Margin of the summer leaf (hyaline/dentate)	BI	Stereo
MAR_WIN_LEAF	Margin of the winter leaf (hyaline/dentate)	BI	Stereo
OUTER_SPI_BRACT_HAIR	Presence of hairs on the outer spikelet bract (yes/no)	BI	Stereo
INNER_SPI_BRACT_APEX	Shape of the inner spikelet bract apex (crenulate/undulate)	BI	Stereo
OUT_SPI_BRACT_APEX	Shape of the outer spikelet bract apex (crenulate/undulate)	BI	Stereo
SUM_LEAF_APEX	Shape of the summer leaf apex (acute/cucullate)	BI	Stereo
VEINS_HAIRS	Presence of hairs along the leaf veins (yes/no)	BI	Stereo
WIN_LEAF_APEX	Shape of the winter leaf apex (acute/cucullate)	BI	Stereo
COL_INV	Involucre colour (green/variegated/reddish/dry)	CN	Stereo
COL_PET	Petal colour (white/pink/fuchsia)	CN	Stereo
GEN_SHAPE_OUT_INV_BRACT	Shape of the outer involucral bract (deltate < triangular < strictly triangular)	CO	Stereo
SHAPE_APEX_INN_INV_BRACT	Shape of the inner involucral bract apex (round < acute < mucronate < apiculate)	CO	Stereo
SHAPE_APEX_INT_INV_BRACT	Shape of the intermediate involucral bract apex (round < acute < mucronate < apiculate)	CO	Stereo
SHAPE_APEX_OUT_INV_BRACT	Shape of the outer involucral bract apex (acute < mucronate < apiculate < acuminated < subulate < long subulate)	CO	Stereo
ANG_SUM_TIP	Summer leaf tip angle (°)	QC	Fiji
ANG_WIN_TIP	Winter leaf tip angle (°)	QC	Fiji
AWN_LENG	Awn length (mm)	QC	Fiji
DIAM_CAP	Capitulum diameter (mm)	QC	Calliper
HEIGTH	Plant height (mm)	QC	Ruler
LENG_CAL _PED	Calyx pedicel length (mm)	QC	Fiji
LENG_CAL_TUBE	Calyx tube length (mm)	QC	Fiji
LENG_INNER_INV_BRACT	Inner involucral bract length (mm)	QC	Calliper
LENG_INNER_SPI_BRACLE	Inner spikelet bracteole length (mm)	QC	Calliper
LENG_INNER_SPI_BRACT	Inner spikelet bract length (mm)	QC	Calliper
LENG_INTER_INV_BRACT	Intermediate involucral bract length (mm)	QC	Calliper
LENG_OUT_INV_BRACT	Outer involucral bract length (mm)	QC	Calliper
LENG_OUTER_SPI_BRACLE	Outer spikelet bracteole length (mm)	QC	Calliper
LENG_OUTER_SPI_BRACT	Outer spikelet bract length (mm)	QC	Calliper
LENG_SUM_LEAF	Summer leaf length (mm)	QC	Ruler
LENG_WIN_LEAF	Winter leaf length (mm)	QC	Ruler
LIMB_LENG	Limb length (mm)	QC	Fiji
SCA_DIAM	Scape diameter at 1 cm from the base (mm)	QC	Calliper
SCA_LENG	Scape length (mm)	QC	Ruler
SHEATH_LENG	Sheath length (mm)	QC	Calliper
WIDTH_CAL_TUBE	Calyx tube width below the limb (mm)	QC	Fiji
WIDTH_IAL_SUM	Width of the hyaline margin in summer leaf (mm)	QC	Fiji
WIDTH_IAL_WIN	Width of the hyaline margin in winter leaf (mm)	QC	Fiji
WIDTH_INNER_INV_BRACT	Inner involucral bract width at the middle (mm)	QC	Calliper
WIDTH_INNER_SPI_ BRACT	Inner spikelet bract width at the middle (mm)	QC	Calliper
WIDTH_INNER_SPI_BRACLE	Inner spikelet bracteole width at the middle (mm)	QC	Calliper
WIDTH_INTER_INV_BRACT	Intermediate involucral bract width at the middle (mm)	QC	Calliper
WIDTH_OUT_INV_BRACT	Outer involucral bract width at the base (mm)	QC	Calliper
WIDTH_OUTER_SPI_ BRACT	Outer spikelet bract length at the middle (mm)	QC	Calliper
WIDTH_OUTER_SPI_BRATLE	Outer spikelet bracteole width at the middle (mm)	QC	Calliper
WIDTH_SUM_LEAF	Width of the summer leaf at the middle (mm)	QC	Calliper
WIDTH_WIN_LEAF	Width of the winter leaf at the middle (mm)	QC	Calliper
N_ WIN_VEINS	Number of veins (with sclerenchyma) of winter leaf	QD	Microscope
N_INV_BRACT	Number of involucral bracts	QD	—
N_SUM_VEINS	Number of veins (with sclerenchyma) of summer leaf	QD	Microscope
SCAP_NUM	Number of scapes	QD	—

**Table 3 biology-11-01060-t003:** Confusion matrix of the LDA based on the 49 morphometric characters, assuming the current taxonomic hypothesis of *Armeria arenaria* subspecies as a priori groups, as proposed by Arrigoni [17]. Rows show the membership of each a priori established group, whereas columns show the membership predicted by the classification model.

*A. arenaria* subsp.	*apennina*	*arenaria*	*marginata*	*praecox*	Total
*apennina*	46	5	8	0	59
*arenaria*	2	105	1	2	110
*marginata*	6	0	28	0	34
*praecox*	0	5	0	15	20
Total	54	115	37	17	223

**Table 4 biology-11-01060-t004:** Confusion matrix of the LDA based on the seed morpho–colorimetric dataset (percentages of correct classification), assuming the current taxonomic hypothesis of *Armeria arenaria* subspecies as a priori groups, as proposed by Arrigoni [17]. Rows show the membership of each a priori established group, whereas columns show the membership predicted by the classification model.

*A. arenaria* subsp.	*apennina*	*marginata*	*arenaria*	*praecox*	Total
*apennina*	36.3	30	15	18.7	100
*marginata*	18.5	71.5	7.5	2.5	100
*arenaria*	15.5	8.2	48.7	27.7	100
*praecox*	20	—	15	65	100

**Table 5 biology-11-01060-t005:** Confusion matrix of the LDA based on the seed morpho–colorimetric dataset (percentages of correct classification), assuming “arenarioid” and “marginatoid” groups for *Armeria arenaria* populations. Rows show the membership of each a priori established group, whereas columns show the membership predicted by the classification model.

Group	Arenarioid	Marginatoid	Total
Arenarioid	87.4	12.6	100
Marginatoid	16.3	83.3	100

**Table 6 biology-11-01060-t006:** Confusion matrix of the LDA based on the seed morpho–colorimetric dataset (percentages of correct classification), according to the alternative grouping hypotheses I and II for *Armeria arenaria* populations. Rows show the membership of each a priori established group, whereas columns show the membership predicted by the classification model.

Groupings		Marginatoid	Arenarioid (LL Excluded)	LL	Total
Grouping hypothesis I	Marginatoid	81.3	12.3	6.5	100
	Arenarioid (LL excluded)	13.1	52.3	34.6	100
	LL	1	21	78.0	100
		**Marginatoid**	**Arenarioid (FB Excluded)**	**FB**	**Total**
Grouping hypothesis II	Marginatoid	81.5	18.5	—	100
	Arenarioid (FB excluded)	13.3	84.1	2.6	100
	FB	—	9	91.0	100

**Table 7 biology-11-01060-t007:** Results of niche similarity tests in environmental spaces among the different taxa and circumscription hypotheses of *Armeria arenaria*. Backgrounds were defined by applying 10 km buffer zones around the occurrence points. Current taxonomic hypothesis as stated by Arrigoni [17], (I) = first alternative grouping hypothesis, (II) = second alternative grouping hypothesis. ns = not significant.

*Armeria arenaria*	subsp.	*arenaria*	*praecox*	*marginata*
Current taxonomic hypothesis	*praecox*	0.036 ^ns/ns^		
	*marginata*	0.041 ^ns/ns^	0.044 ^ns/ns^	
	*apennina*	0.006 ^ns/ns^	0.000 ^ns/ns^	0.208 ^ns/ns^
Grouping hypothesis I	*praecox*	0.036 ^ns/ns^		
	*marginata*	0.033 ^ns/ns^	0.012 ^ns/ns^	
Grouping hypothesis II	*praecox*	0.005 ^ns/ns^		
	*marginata*	0.000 ^ns/ns^	0.107 ^ns/ns^	

## Data Availability

The data presented in the current study are available within the article and Appendix A.

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
