# Peer review of "Integrative Taxonomy of Armeria arenaria (Plumbaginaceae), with a Special Focus on the Putative Subspecies Endemic to the Apennines"

_biology, 2022, doi:10.3390/biology11071060_

Round 1

Reviewer 1 Report

The Authors of the manuscript under review have revised the current knowledge of the Armeria arenaria taxonomy, and especially the putative subspecies considered endemic to the Apennines.

In addition to morphometric analyzes, a number of modern methods were used in the research, such as seed morpho-colorimetry, karyotype asymmetry estimation, molecular systematics, and comparative niche analysis.

The Authors showed that A. arenaria populations in northern Italy and the central-northern Apennines are different. In the first area, only the presence of A. arenaria ssp. praecox was confirmed and population of A. arenaria ssp. arenaria was not distinguished. Populations occurring in the Apennines were included in only one subspecies of A. arenaria ssp. marginata, while A. arenaria ssp. apenina, the presence of which was reported by other authors, was not confirmed.

I did not notice any errors in the manuscript, except for Table 1. The first column of the table is unreadable - code numbers overlap with the acronyms of localities (maybe only my Adobe Acrobat shows it like that). Moreover, all abbreviations used in the table should be explained in the caption of the table, not only in the text of the article.

Author Response

Dear Reviewer 1 

We are deeply grateful for your comments and suggestion. We modified our ms accordingly to your suggestions. You can find more comments the in the file attached.

Sincerely,
The Authors

Reviewer 2 Report

Dear authors and editors,

The research has been carried out carefully and thoroughly. It represents an important contribution to the knowledge of European flora. I suggest you accept it for publication in this journal.

All chapters are written clearly, and supplements contain meaningful information.

The only detail I found unclear is concerning the confusion matrices. For me, it is not clear what are the original and revised determinations. Please indicate this in the title row and title column of each matrix.

I wish you all the best.

Author Response

Dear Reviewer 2

We are deeply grateful for your comments and suggestion. We are also glad that you appreciated our work. We modified our ms accordingly to your suggestion. You can find more comments in the file attached.

Sincerely,
The Authors

Reviewer 3 Report

The authors conducted analyses of Armeria arenaria s.l. using different sources of evidence focused on Italian populations. This include morphometrics (using 49 characters), imaged-based morphometrics of fruits, karyological analyses, phylogenetic analysis based on Sanger DNA sequences (ITS and ptDNA), as well as environmental niche comparative analyses. Elaborating on the results of these analyses, they propose taxonomic changes that include rejecting the presence of A. arenaria s.str. in Italy, referring Italian populations to subsp. praecox and recognizing a single Apennine subspecies in the complex, subsp. marginata.

The results appear to be sound for the most part and the proposed integrated taxonomy approach is particularly appropriate in a genus that exhibits large taxonomic problems due to weak reproductive isolation barriers and extended hybridization. But such taxonomic difficulties demand special care when analyzing and integrating data. I think the ms. can be published, but the authors need to provide further information on several aspects, clarify some points and address some concerns, without which the reader cannot have certainty of the conclusiveness or even the soundness of the results, which is why I recommend 'major rev.' The manuscript is well-written in general, but still needs some editing mainly regarding figures.

In my opinion, the main issue in need of revision is the phylogenetic analyses. In a genus in which ITS variation exhibits a striking geographic structure of variation derived from extensive gene flow which partly overrides the taxonomic structure, more information must be provided, at least under supplementary material, on polymorphisms and patterns of variation for this marker. In Fuertes and Nieto Feliner (2003), the four sampled Italian accessions, all from southern part of the country, fall in the same clade.  Another example: Baumel et al. 2009, which the authors cite, detected numerous polymorphisms for ITS informative sites in French populations. With the ample report in the literature of intragenomic ITS variation resulting from hybridization (Fuertes & Nieto Feliner 2003; Nieto Feliner et al. 2004;…), some information and comment on the variation in this multigene marker would be most appropriate, more so when presumably they are using direct sequences where intragenomic polymorphism is not always evident. Along the same line, just reporting that the ILD test is negative is not enough information. This test has been criticized for increasing the probability of type I errors, which incorrectly reject the correct hypothesis of congruence (Darlu & Lecointre, 2002; Dowton & Austin, 2002…). I do not say you shouldn’t perform this test, but to clarify these issues, more details on analyses and results should be provided (perhaps under supplementary material). For instance, the probability of congruence between each pairwise comparison considering separately each of the plastid regions, and the significant level. Even if they convince the reader on the combinability of the ITS and ptDNA datasets, due to the same reasons argued above, the authors should run and show independent ITS and plastid DNA trees in addition to the combined ones, supplementary material.  Otherwise the reader is missing information in a genus where hybridization is frequent. Further, the reader does not even know how many of those 36 informative characters correspond to each of the markers, which is relevant when comparing and integrating topologies.

Another concern is the terminology, which should be revised. First, there are several terms (in Table 2, identification key, and elsewhere) that apparently refer to bracts but are very confusing: “inner bracts” (which ones?); scales; spikelet bracts; “bracts of the inner spikelet” (L. 550); etc. I think the term “scale” may come from Arrigoni (2015), but it seriously compromises understanding. In the literature of the last decades “involucral bracts” denote those integrating the involucre, “spikelet bracts” denote the ones embracing the spikelets inside the glomerule (absent in some SW Iberian taxa), and, although not always described, there might also be “bracteoles” inside the spikelets. Please adapt consistently your descriptions to this scheme. In addition, it is not clear to me what spine (on the calyx awn) is. Do you mean “scabrous” awn? In addition, you refer to ‘seeds’ repeatedly (abstract, lines 94, 158,…) even in a heading (L. 213) and profusely under it. Does this mean you are taking images of the seed after removing the fruiting calyx and without the membranous pericarp? I would not expect differences in this plain organ whereas there are plenty of useful taxonomic differences across the genus when you observe the fruiting calyx with its seed included. I cannot imagine that 124 morphometric characters could be identified in a nude seed. But please clarify what you are measuring.

My last general comment concerns the two-letter acronyms referring to the populations sampled, used throughout the manuscript, sometimes without enough contextual information to help the reader understand what you want to say. This is particularly obvious when you are referring to a figure which ideally should clarify things, but since the figures all refer to Table 1 for acronyms, the reader gets easily lost. I am sure you associate acronym right away taxon and geography. But this is not the case for the reader. See below my specific comments particularly on the figures.

Other comments

Title: Since only two of the samples are non-Italian, instead of “with a Special Focus on the Putative Subspecies Endemic to the Apennines”, I suggest “in Italy” or “with Focus on the Apennines”.

L. 51.- “laid” instead of “placed”

L. 63.- If you’re referring to Popper’s criterion, I would say “falsifying” instead of “confuting”, which I have never heard.

L. 106-107.- You use the term “current taxonomic hypothesis” repeatedly throughout the ms. I agree that it is a good practice to consider taxonomic treatments resulting from phylogenetic analyses as hypotheses because they can be tested with more data and analyses. But, the hypothesis you are referring to (Arrigoni 2015) is not based on phylogenetic data and so there’s no direct formalized way that new characters would prove that hypothesis incorrect. I suggest (only a suggestion) ‘current taxonomic treatment’.

L. 197-200.- For clarity, I suggest that instead of using “respectively”, you rephrase.

L. 231.- What do you mean with an “statistical elaboration”?

L. 345.- “surges”. Do your mean “raises”?

L. 353.- Here in the morphometric analyses and elsewhere (e.g. , in captions of figures 3, 4 and headings of Tables 6, 7, …) you establish two groupings that are useful for addressing the questions you are discussing. In the Seed section you name them (I and II). Please use consistently this or other numbering throughout the ms. because it will help the reader.

L.456-464.- It is very difficult to follow these sentences without adding some labelling to Fig. 5. I suggest not only some taxonomic (marginatoid…) but also geographic.

L.462-464.- The “second main clade” is not “subdivided into two clades” because PS does not fall into any of those two. You may say that it “contains two clades”.

L. 507-513.- Very confuse (and partly conceptually wrong) sentences. Incomplete lineage sorting is not restricted to, or detectable in, recent diversification. More on the contrary. It is rather manifested in the medium and long term. Then you mention “the admixture of the PS…” The idea that this population is introgressed sound reasonable, but I don’t see where it has been mentioned before that it shows admixture. So, it seems that the two potential causes for the phylogenetic placement of the PS population are mentioned in a rather contradictory way. Please revise.

Fig. 3.- It would help to add on the axes the % of variance explained by each of the LD functions

Fig. 4.- About the polygons, if they were ellipses enclosing the points you need not explain anything, but since these polygons are formed connecting actual points, some explanation on the caption may be appropriate.

Fig. 5.- This figure is poorly informative unless you add some extra labeling and/or overlaying information. Even just identifying the two groups mentioned under results require investing some time. I also suggest that you move upwards the posterior probability PP (1) of the lowest clade so that it falls on the branch, not on the polytomy.

Fig. 6.- Please add on the map the two-letter acronyms that you repeatedly use across the ms.

Author Response

Dear Reviewer 3

We are deeply grateful for your precious and meticulous comments and suggestions that improved our work a lot. We modified our ms accordingly to your suggestions. You can find more comments the in the file attached.

Sincerely,
The Authors

Round 2

Reviewer 3 Report

Thank your for considering the points I raised. The ms. has improved substantially.